# Fostering Green Entrepreneurship and Women's Empowerment through Education and Banks' Investments in Tourism: Evidence from Serbia

**Mirjana Radović-Marković [1,2,\*] and Branko Živanović [3]**

[1]  Department of Basic research, Institute of Economic Sciences, 11000 Belgrade, Serbia
[2]  School of Economics & Management; South Ural State University, 454080 Chelyabinsk, Russia
[3]  Department of Finance and banking, Belgrade Banking Academy, 11000 Belgrade, Serbia;
    branko.zivanovic@bba.edu.rs
\*  Correspondence: mradovic@gmail.com

**Abstract:** The aim of our research is to consider the potential for women's empowerment through tourism and women's equality inherent in the green economy. In addition, our research should shed more light on the women's dimensions of green growth, especially in the context of development of entrepreneurship in tourism. In line with this, our approach in the study combines a women's perspective with green growth and entrepreneurship development in the tourism sector in Serbia. The research was carried out in the most important tourist centers in the country, such as Novi Sad, Nis, Zlatibor, Vrnjačka Banja, and Sokobanja. This study showed that insufficient attention has been dedicated to this industry from the perspective of its benefits for women. In addition, the research indicated that, in the field of tourism, women mostly prefer special programs of education that are adjusted to the job requirements that they have already been performing or to a similar job that they are just about to start. It is necessary to involve them more often in various projects that encourage their further empowerment. The research also discovered gaps in the supply of finance between the needs of green entrepreneurs in tourism and what the financial system is willing to provide to them. Firstly, there is a lack of appropriate lending products offered by the commercial banking sector. In particular, a combination of financial support and suitable financial tools to encourage women's initiatives for start-ups in tourism is missing.

**Keywords:** women's empowerment; inclusive tourism; sustainable development goals; investments

**JEL Classification:** J11; J21; J22; J24; J43

## 1. Introduction

A green economy can generate employment and new business opportunities in various sectors. In this context, "green growth is a consequence of and a means of greening the economy" [1] (p. 9). In line with this, the green economy must coexist with other sustainable development concepts. Sustainable tourism development should meet the needs of present tourists and host regions while protecting the environment of any country [2]. However, it is often difficult to predict the influence of seasonal tourism in each environment [3]. In this context, it is necessary to minimize the environmental impact and to maximize the socioeconomic advantages of tourist destinations. Therefore, it is important to make a balance between sustainability and a country's economic benefits [4].

The transition to a green economy will vary from one country to another, as it depends on the specifics of each country's natural and human capital and on its relative level of development [5]. Sustainable and responsible tourism development is not possible without the application of ecological thinking.

We considered tourism in our study as a human activity that is closely linked with human behavior in interaction with other people, economies, and environments [6]. Tourism was selected for research since it plays a central role in the economic plan in many countries of the world, especially in Serbia and other Western Balkan countries. Thus, it is expected that tourism will continue to develop in the near future and make a larger contribution to GDP than the 3.2 percent made in 2018 [7]. Such a trend in the development of tourism opens up great opportunities for opening new jobs through the promotion of entrepreneurship in tourism, where women should see their chance [8]. Thereby, tourism influences the protection of the environment and strengthening of the labor market through the employment of a large number of workers [9]. In the period from 2012 to 2017, the number of tourists in Serbia increased from 922,000 to 1,470,000 tourists, and from the total number of tourists who visited Serbia in 2017, about 90% were tourists from Europe [10]. This growth of tourism reflects on the environment in various ways. According to numerous studies, the growth of tourism and environmental protection should not be seen as opposing forces [11–13]. However, the relationship of tourism with the environment is complex because tourism involves many activities that can have adverse environmental effects. This impact can be negative and positive in the sense that it may trigger different types of responses. A negative effect can influence economic, sociocultural, and environmental responses in a region or country as follows:

(a)    Loss of biodiversity that can be attributed to tourism, or
(b)    air pollution caused by tourism-related transport.

When we consider business opportunities in a green economy, we can point out that entrepreneurship has the potential to be a catalyst for a positive change in both the economic and environmental spheres [14]. In many countries, including Serbia, more attention has been paid to profit creation than to environmental protection in the context of tourist offers. Managing these impacts is a core function of the investment promotion process outlined here [15]. Namely, investment and finance have important roles to play in supporting inclusive tourism and female entrepreneurship development. A country's investment is influenced by many factors, including the laws and regulations that impact access to capital and credit, as well as both foreign and domestic investors in tourism-related businesses. Accordingly, the study will explore whether there are obstacles to obtaining funding to start a tourism business.

The World Tourism Organization (UNWTO) study [8], which analyzed women's positions in tourism, led to the following conclusions:

1.    Women make up a large part of the formal tourism workforce.
2.    Women earn up to 15% less than the male population.
3.    There were five female ministers of tourism in the world in 2010.
4.    Women in the parent companies mostly perform unpaid jobs. However, as in other sectors of the economy, and even in tourism, there has been a division of jobs between men and women.

Integrating gender perspectives into the discussion of tourism is particularly important, as the tourism industry is a major employer of women, offers various opportunities for income-generating activities, and at the same time, affects women's standards of living. Therefore, the motivation for this research comes from the lack of investigation of the role of women in the tourism industry. In addition, the study will explore the relationship between education and entrepreneurial activities in tourism among women.

In addition, the aim of our research is to analyze objectives held by individuals seeking a transition to a green economy:

i     Environmental sustainability

ii    Employment and business opportunities in tourism with a focus on women

In the theoretical part of the research, the question of forming a unique concept of empowerment of women through tourism is raised, and various contemporary approaches to the topic are presented.

## 2. Hypotheses

Entrepreneurship education can awaken entrepreneurial spirit and can foster a positive attitude towards risk-taking and learning from failure for females engaged in entrepreneurial activities, resulting in a boost for other females who intend to participate in same. In the past decades, education programs in tourism are almost always linked to higher educational institutions worldwide. However, these programs tend to use more traditional and theoretical approaches than practical work. A number of researchers share the same opinion: That formal higher education does not contribute to increasing the competencies [4,16]. Namely, there is an opinion that women with no instruction concur with this idea more so than the ones with advanced education [6]. On the other hand, one group of scientists thinks that formal qualifications are necessary, but not always required in tourism [17]. In addition, there is little documented evidence demonstrating which specific factors within the curricula are effective in fostering the entrepreneurial abilities of students through education and raising entrepreneurial intentions after the students' graduation. In this context, there is still a dilemma regarding which form of education contributes to competency-based learning. Accordingly, we have tried to test these different settings in our paper.

**Hypothesis 0 (H0).** *Formal education (university degree) does not affect the enhancement of competences in tourism (Arranz, Ubierna, Arroyabe, Perez, and Arroyabe, J.C., 2017) [16].*

**Hypothesis 1 (H1).** *The level of education of the respondents directly determines the attitude in which competences are best enhanced (Brennan, Chanfreau, Finnegan, Griggs, Kiss, and Park, 2015) [17].*

## 3. Theoretical Overview

Numerous schools exist that address the issues of green economy and sustainable development [18–21]. "The initiative to green the economy shows that greening the economy is a new driver of growth that generates new jobs and contributes to poverty reduction ([22] p. 4; [23]). According to our opinion, many researchers in this area started from common premises. However, they came to a strong disagreement with respect to the effects of greening the economy on economic growth. Furthermore, it has been shown that it is necessary that economic research is linked with the research in the fields of ecology and environmental protection in order to anticipate and mitigate the effects of climate change, soil degradation, greenhouse gas emissions, and anything that is a threat to the future and survival of the global population. The goal of a green economy is to foster economic development, employment growth, and an increase in earnings while taking care of the prevention of environmental catastrophes and other externalities [24].

"Green Economy promotes a triple bottom line: Sustaining and advancing economic, environmental, and social well-being" ([24] p. 2). In other words, progress towards sustainability is made in terms of the ecological, social, and economic well-being of the community and country as a whole ([25] p. 417). Recent research showed that the sustainable development is usually based on the three pillars: Environmental, sociocultural, and economic. All three pillars (3Ps) are relevant for developing green policies for the economy in any country [26].

Over the past decade, the green economy has played an important policy framework for sustainable development [27,28]. This policy framework needs the distinctions between 'green economy', 'green growth', and 'green recovery'. However, the term 'green economy' is not consistently defined. Consequently, different schools of thought on sustainability and green economy have emerged [29,30].

According to all definitions, "green economy is one that is environmentally sustainable in the broadest sense; that is, it is an economy that operates without infringing environmental limits" ([31] p. 5). On the other side, the term "green growth describes an economic growth strategy based on the ecological restructuring of existing economic processes, creating jobs and income generation opportunities in new 'green' sectors of the economy, and minimizing environmental impacts" [32]. In addition, sustainable entrepreneurship plays an important role in solving the employment problems and promoting sustainable social and economic development in an environmental context [33].

Since there is no unique definition of green economy nor of the strategy of sustainable growth, the variety of economic policies can be considered according to the green growth, ranging from those that deem it a priority goal to those that do not focus upon it at all [34].

The three main areas for the current work on Green Economy are as follows [35]:

(1)　Promoting macro-economic sustainable economic growth.
(2)　Demonstration of Green Economy approaches with a focus on access to green finance, technology, and investments.
(3)　Supporting the transition of countries to Green Economies.

According to some scholars, green economy covers the achievements of the sustainable development concept, expanded by the desire to improve human well-being and social and gender equality [36]. Although there is no clear relationship between gender equality and sustainability, both make an impact on economic development [37]. However, there is a bidirectional relationship between economic development and women's empowerment, which is characterized as enhancing the capacity of women to access the constituents of advancement, specifically well-being, education, earning opportunities, rights, and political participation [37] (p. 13). That is, through green entrepreneurship, women can gain status, approval, and recognition in society [38]. Therefore, many scholars do agree that green entrepreneurship enhances women's self-realization and strengthens their position in society [39].

Tourism development is consequently reflected in the empowerment of women in a global business environment [40,41]. Namely, tourism presents both opportunities for gender equality and women's empowerment [42,43]. In this context, gender-sensitive policies are of key importance to ensure that women and men can equally benefit from a green(er) economy [1]. With a transition to the green economy, numerous green(er) jobs will be created. However, existing gender differences in education do not correspond with the skills needed in the emerging green economy, constituting another obstacle [1,19,20]. According to The Organisation for Economic Co-operation and Development (OECD) research [1], the skills and professions identified as particularly relevant for the green economy tend to be male dominated (p. 6). For the tourism sectors, it is critical to consider factors such as educational elements and attitudes of people toward green tourism [44].

Therefore, training programs should be a part of a long-term strategy to match women's skills to the needs of the tourism industry [36]. For them, training should include the acquisition of new knowledge in the following fields:

(a)　Ecological environment protection in tourism
(b)　Relevant laws and policies
(c)　Effective promotion and branding strategies
(d)　Green tourism intention and behaviors
(e)　Effective marketing for the destination
(f)　Understanding tourism planning
(g)　Effective decision-making
(h)　Management in tourism
(i)　Best practices within tourism types and similar.

Strengthening environmental protection education and promotion of the positive impact of tourism development improves the ability for environmental protection [10]. In addition, higher education and bachelor curricula must therefore bring together all of these concerns and be able to promote entrepreneurial spirit in the future professionals. People with entrepreneurial spirit and with great management skills are able to take risks and allow organizations to be sustainable in financial, social, and economic senses, both in tourism and on the organizational level [36].

"The development of an open educational program is often initiated by a strong will to provide an alternative means of professional and personal development" ([45] p. 15). For these reasons, among other things, this research investigates the appropriate training programs for women seeking employment with a focus on the tourism industry.

In addition, access to financing is of key importance to promoting entrepreneurship and Small and medium-sized enterprises (SME) development, as well as building a competitive, innovative, and sustainable tourism sector. However, tourism SMEs have different financing needs and face different challenges at each stage of the business lifecycle [46]. Wide ranges of private and public financing instruments are necessary to create and develop the businesses and to remain competitive.

## 4. The Conceptual Framework of the Study

The conceptual framework which was developed for this study is shown in Figure 1. Figure 1 explains the dependent and independent variables, where socioeconomic factors, competency-based learning, access to financial services adapted to tourism business needs, government policies, and development programs are independent variables that may affect the engagement of women in the tourism sector. Consequently, women's empowerment is directly affected by these independent factors.

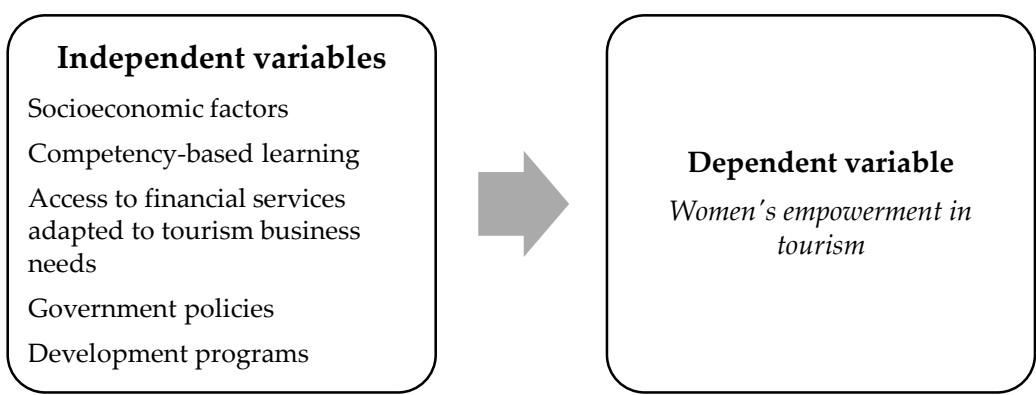

**Figure 1.** A simple conceptual framework for the study. Source: Authors.

Bearing in mind that each country has its own visions and opportunities for women's empowerment, development programs and their implementations will vary by country. Hence, the results achieved will also be different. Accordingly, starting and running a business in tourism can vary according to national and geopolitical factors, as well as industrial, governmental, and cultural factors. These factors may be considered when we explain difficulties encountered by those who create policies for favoring development of new green businesses and entrepreneurial activities in tourism.

## 5. Methods of the Study

The present research was conducted in the Republic of Serbia, which is one of the Western Balkan countries. The country has huge potential for health tourism—it has over 1000 cold and warm mineral water springs, plus a wealth of natural mineral gases and medicinal mud. The country also has potential for development of different types of tourism—rural tourism, cultural tourism, old craft tourism, sports tourism, gastronomy tourism, and others. The location is presented below in Figure 2.

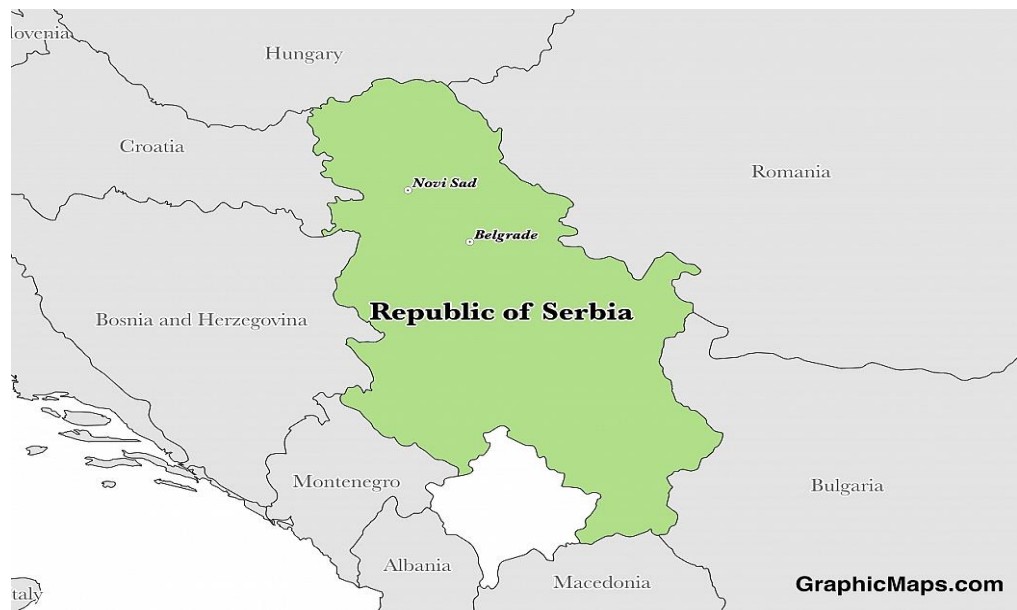

**Figure 2.** Location of the study area.

The methodology of our paper is based on qualitative and quantitative research. Namely, we used the methods of analysis and synthesis as well as the method of deduction for the interpretation of the obtained data. The Kolmogorov Smirnov, Lilliefors, and Chi-Square Tests were used. Also, in our research, the "Contingency Coefficient" and the "Cramer's V Coefficient" were applied.

At the end of the year 2018, the research was started. Data were collected from 120 respondents from December 2018 to June 2019. The largest group of respondents was that of the age group between 26 and 34 (30%), while the smallest group was the age group from 18 to 25.

The research was carried out in the most important tourist centers in the country, such as Novi Sad, Nis, Zlatibor, Vrnjačka Banja, and Sokobanja.

*Data Collecting Procedure and Statistical Technique*

In this study, data were collected using multiple methods, including a literature review which used material from tourism organizations and the ministry responsible for tourism in Serbia, direct interviews, and a random sample survey. However, the research was conducted primarily through the interviews and the survey.

The interviewer selected the respondents based on their availability. Before the interview, the researchers described the meaning of the study and its context to the participant. We used open-ended and closed-ended questions to explore the factors that could influence women's empowerment through tourism.

The interview and survey questions were: "Do you have formal education in tourism?", "Would you like to establish a business in tourism, and what type of business do you prefer?", "What factors can influence your decision to start up a business in tourism?", "What obstacles are women faced with when starting a business in tourism?", and "What are the opportunities for strengthening the position of women in tourism?" (Table 1).

Each interview took about 15 min. All participants were interviewed on a one-to-one basis.

After completion of the interview and survey, the collected data were coded, tabulated, and analyzed in accordance with the objectives of the study.

Descriptive techniques were used to analyze the collected data by using the concerned software, such as SPSS and Microsoft Excel.

**Table 1.** Research methods and design.

| Stage | Research Objectives | Research Questions | Information Source |
|---|---|---|---|
| Before the research | Construct theoretical framework | Hypothesis and aim of the research formulation | Information from relevant literature |
| | Collect basic information of the sample of tourist places | Gathering information | Archival documentation |
| During the research | Collect data | "Do you have formal education in tourism?" | Interview and survey |
| | | "Would you like to establish a business in tourism and what type of business do you prefer?" | Interview and survey |
| After the research | Data collected were coded, tabulated, and analyzed in accordance with the objectives of the study. | "What factors can influence your decision to start up a business in tourism?" | Interview and survey |
| | Descriptive techniques were followed to analyze the collected data | "What obstacles are women faced with when starting a business in tourism?" | Interview and survey |
| | The results are presented | "Does the bank provide credit and loans to companies in the tourism sector?" | Interview |
| | | "Is there a difference in criteria when considering a loan request for different types of tourist activities?" | Interview |
| | | "Do you have special credit for women entrepreneurs and criteria for the provision of such credit and/or services?" | |
| | | "How much credit the bank did approve for women in the period between 2016 and 2019, and what type of credit was it?" | |
| | | "What are the opportunities for strengthening the position of women in tourism?" | Interview and survey |

Source: Authors.

Descriptive analyses, such as range, number, percentage, mean, standard deviation, and rank order, were used whenever possible. Pearson's product moment coefficient of correlation (r) was used in order to explore the relationship between the concerned variables. Throughout the study, a level of probability of at least a 5% (0.05) was used as the basis for rejecting a null hypothesis.

The data collected in the first round were reviewed by the authors individually. In the second round, authors discussed individual findings and agreed on group findings. The results are presented by a method comparing the attitudes of women aged from 18 to 65 and their different levels of education relating to the opportunities and barriers within entrepreneurship activities in tourism.

## 6. Key Findings and Discussion

The analysis based on the respondents' answers to the question "Do you have formal education in tourism?" showed that 90% of all respondents have education in tourism (Figure 3).

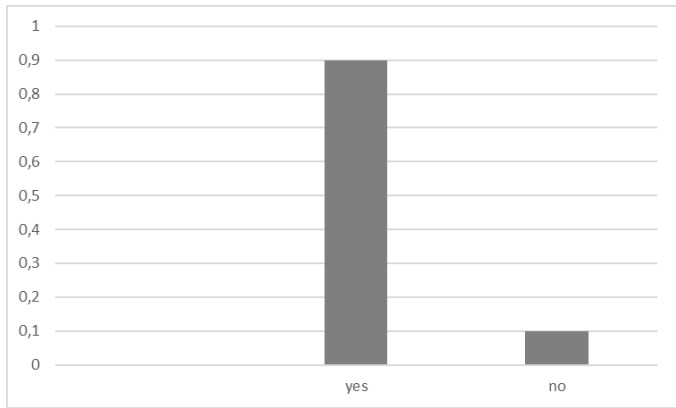

**Figure 3.** Percent of females with formal education in tourism. Source: Authors.

Female respondents were asked if they would establish a business in tourism and if they preferred to be involved in cultural tourism (39%), and the smallest proportion of them chose the manufacturing of souvenirs and different crafts as their prospective business (8.8%).

The research showed that the largest proportion of female respondents in all age groups is willing to invest in business start-ups in tourism (76%). However, they believe that their financial resources are mostly insufficient for that. Therefore, the biggest problem they face when starting a business in tourism is a lack of financial resources (50%) (Figure 4).

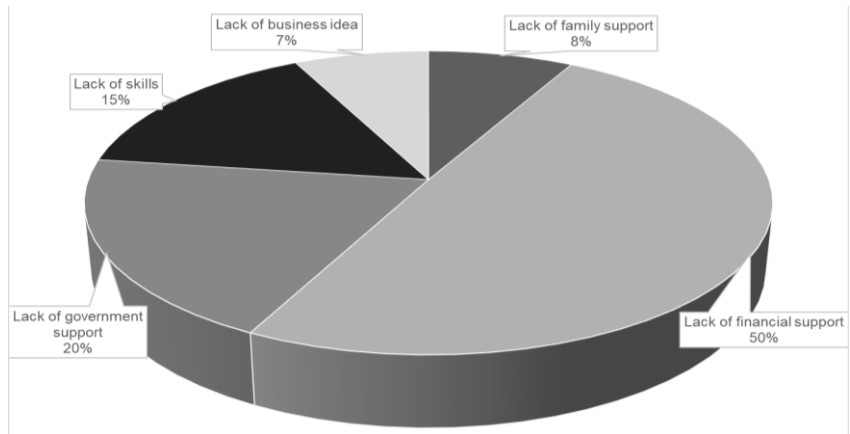

**Figure 4.** Obstacles women face when starting a business in tourism. Source: Authors.

The cause may be found in the lack of appropriate lending products offered by the commercial banking sector due to the small size and high risk associated with microcredit borrowers. However, to find out the exact reason, we did additional research. Accordingly, an interview was conducted with the bank executives; we asked them the following questions:

1. Does the bank provide credit and loans to companies in the tourism sector?
2. What type of loans and credit do you offer in tourism and under what conditions?
3. Is there a difference in criteria when considering a loan request for different types of tourist activities (catering, hotel and restaurant, tourist agencies, transport, retail, service, tourist organizations)?
4. Do you have special loans for women entrepreneurs, and what are the criteria for the provision of such credit and/or services?
5. How many loans did the bank approve to women in the period between 2016 and 2019, and what types of loans were they?

A total of 22 banks of the 26 in the banking system of Serbia participated in the questionnaire in 2019. However, 17 banks (77.27%) answered our questions.

Answers to our first question showed that 94.12% of banks provides credit and loans to companies in the tourism sector (Table 2). However, there is no evidence that banks approved the placement.

**Table 2.** Credit and loans to companies in the tourism sector.

| Question | Answers | |
|---|---|---|
| | (a) Yes | (b) No |
| Does the bank provide credit and loans to companies in the tourism sector? | 94.12% | 5.88% |

Source: Authors.

To the second question, "What type of loan products does your bank offer in tourism companies?", all banks replied that they offer long-term and short-term loans.

The reply to the third question, "Is there a difference in criteria when considering a loan request for different types of tourist activities (catering, hotel and restaurant, tourist agencies, transport, retail, service, tourist organizations)?", showed differences in 64.71% of respondents.

When banks were asked if there are special credit products for women entrepreneurs in the field of tourism and what the criteria for their approval are, 94.12% of the answers were that there are no special credit lines for them (Table 3).

**Table 3.** Special credit products for women entrepreneurs in the field of tourism.

| Question | Answers | |
|---|---|---|
| | (a) Yes | (b) No |
| Are there any special credit products for women entrepreneurs in the field of tourism and what are the criteria for their approval? | 5.88% | 94.12% |

Source: Authors.

The answers to question five, "How many loans the banks did approve to women in the period between 2016 and 2019, and what types of loans were they?", showed that only 5.88% of respondents claimed that up to 100 loans were granted to women owners, while only 10 were granted for investments in tourism. Accordingly, our research has shown that urgent financial support to women is necessary in tourism or to other entrepreneurs in the form of subsidies, issued guarantees, tax incentives, soft loans, or similar means of financial help.

According to the Serbian evidence, it is not only the minority that lacks access to efficient and reliable financial services at affordable interest rates. In looking to the future, it would be necessary to try to dispel microfinance "myths" and revisit ongoing debates in microfinance (particularly about how it works, which customers can be profitably served, and what the appropriate role of subsidies is), setting out ideas that could help evaluate experiences to date, frame debates, point to new directions and challenges, and improve the relationship between commercial banks, the SME sector, and entrepreneurs [47].

Similar questions were sent to 30 banks in Serbia in 2016. "The banks were asked to submit information on the credit offered, on whether they had special credit aimed at entrepreneurs, as well as what the conditions for receiving such credits were, especially what types of collaterals they accepted" ([48] p. 63). The results showed that banks did not have any special products or services targeting women in 2016 or in 2019. Therefore, there were no changes over the observed period of three years.

## 7. Opportunities for Strengthening the Position of Women in Tourism

Most respondents almost equally prefer informal education (19%) and special programs (22%), as well as acquiring knowledge through practical work and experience (53%) (Figure 5).

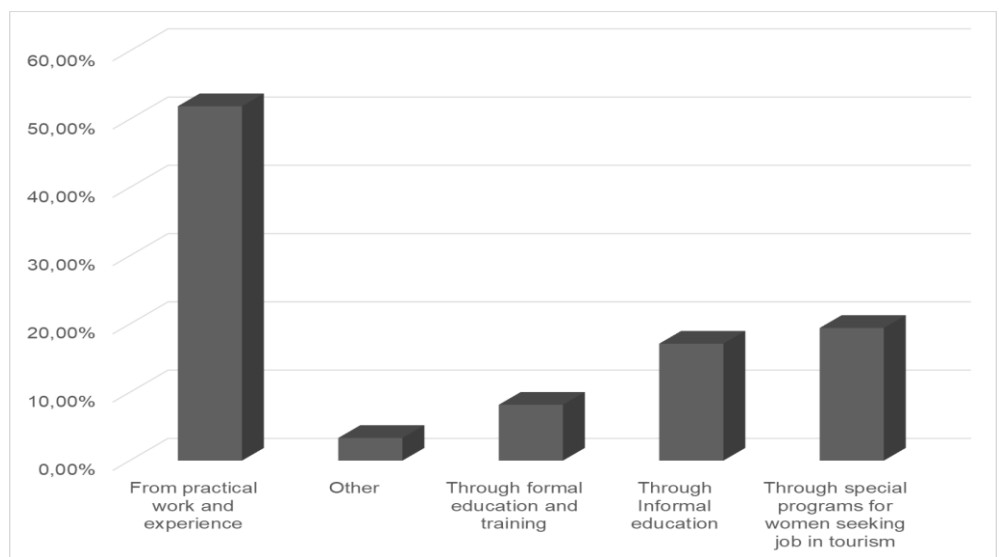

**Figure 5.** Opportunities for strengthening the position of women in tourism. Source: Authors.

The general opinion is that these programs cannot be separated. In addition, e-learning programs specially designed for entrepreneurs can be used as a tool for empowering female competencies in this area. It is especially important in countries with an inadequate system of formal education, like Serbia [49].

As an answer to the question of why it is important to empower women through tourism, our female respondents indicated that the most important reason is the acquiring of gender equality (36.4%). A lower number of respondents had at least some knowledge about the programs relating to empowerment of women (41.6%). Most of them had never participated in any of the projects on this topic (85%).

*Testing of the Hypotheses*

For testing of the hypotheses, we must first examine the data used to determine that the data follow a normal distribution. Testing the hypotheses is done by using the Lilliefors Normality Distribution Test (based on the Kolmogorov–Smirnov Nonparametric Test).

The initial hypotheses for normality testing are as follows:

**Hypothesis 0 (H0).** *The distribution of variable values falls under the normal distribution.*

**Hypothesis 1 (H1).** *The distribution of variable values does not fall under the normal distribution.*

Considering that a significant level of variables is lower than 0.05, the H0 hypothesis cannot be confirmed, while the H1 hypothesis is accepted (Table 4).

**Table 4.** Kolmogorov-Smirnov Test.

| Variable | Significance Level | Result |
| --- | --- | --- |
| Country | 0.000 | Rejected H0 |
| Highest level of formal education | 0.000 | Rejected H0 |
| What is the best way to increase skills? | 0.06 | Confirmed H1 |

Source: Authors.

Because the H1 hypothesis is accepted, the further testing can be carried out on a non-parametric principle.

The One-Sample *t*-Test was used to test the impact of formal education in decision-making as the most influential method for enhancing tourism competences (Table 5).

**Table 5.** One-Sample *t*-Test.

| | | t | df | Sig. (2-tailed) | Mean Difference | 95% Confidence Interval of the Difference | |
|---|---|---|---|---|---|---|---|
| | | | | | | Lower | Upper |
| What is the best way to increase skills? | Through formal education and training | 43.736 | 392 | 0.000 | 1.84733 | 1.7643 | 1.9304 |
| | Through special programs for women seeking jobs in tourism | 20.061 | 392 | 0.000 | 0.84733 | 0.7643 | 0.9304 |
| | From practical work and experience | −3.615 | 392 | 0.000 | −0.15267 | −0.2357 | −696 |
| | Through informal education | −27.290 | 392 | 0.000 | −1.15267 | −1.2357 | −1.0696 |

Source: Authors.

It could be noted from Table 5 that, regarding the significance level, none of the given answers are significant for the level of the average value differences. However, regarding the value of students' distribution (t-level), it is noted that the highest value is for "Through formal education and training", which leads to the conclusion that its influence is of minor significance. Therefore, the H0 Hypothesis has not been confirmed.

The basic question to be answered concerns the impact of individual independent variables on the survey results. In this case, the impact of two variables on the results of the survey was tested.

The test of the independence of (categorical) variables, i.e., "Chi-Square Tests", is used for testing the null hypothesis of two variables that should show mutual independence.

Results of the set versions of hypotheses 1 and 2 are presented in Table 6.

**Table 6.** Results of the set versions of hypotheses 1 and 2.

| | |
|---|---|
| 1 | H0: Respondent's country of origin and "What is the best way to increase skills" are mutually independent variables |
| | H1: Respondent's country of origin and "What is the best way to increase skills" are mutually dependent variables |
| 2 | H0: Respondent's education level and "What is the best way to increase skills" are mutually independent variables |
| | H1: Respondent's education level and "What is the best way to increase skills" are mutually dependent variables |

Source: Authors.

In order to examine the relationship between these two variables, Pearson's Contingency Coefficient and Cramer's Coefficient were calculated as well (Tables 7 and 8).

**Table 7.** Chi-Square Tests.

| Hypothesis | | Dimension | Pearson Chi-Square | Asymptotic Significance (2-sided) | Result |
|---|---|---|---|---|---|
| 1 | What is the best way to increase skills? | Country | 27.151 | 0.000 | Rejected H0 |
| 2 | | Highest level of formal education | 18.042 | 0.006 | Rejected H0 |

Source: Authors.

**Table 8.** Symmetric Measures.

| Hypothesis | | Dimension | Contingency Coefficient | Cramer's V | Result |
|---|---|---|---|---|---|
| 1 | What is the best way to increase skills? | Country | 0.255 | 0.186 | Weak dependency |
| 2 | | Highest level of formal education | 0.210 | 0.152 | Weak dependency |

Source: Authors.

Since the significance value levels in both cases are <0.05, the H0 hypothesis is rejected.

The hypothesis H1 is not confirmed, because our study showed that the level of education of the respondents does not directly determine the relationship regarding the best ways to improve competences in tourism (weak dependency).

## 8. Conclusions

Despite certain improvements over the last five years, the SME development in tourism does not have satisfactory results in terms of development of green business in Serbia. Therefore, the sector of tourism continues to hold a great potential for employment of women and generating income, thus achieving ways to eliminate women's poverty and improve their economic and social position.

Our research showed that this potential is underutilized for women's empowerment in Serbia, even though women are interested in engaging in this industry. Namely, this research indicated that the largest proportion of female respondents in all age groups is willing to invest in business start-ups in tourism. However, the biggest problem they face when starting a business in tourism is a lack of financial resources, followed by a lack of appropriate skills. Therefore, the training and education of women in tourism are extremely important for maintaining of their competitiveness of their business within the tourism industry [50]. For this reason, tourism programs need to be planned and monitored carefully using a sustainable approach. They should include acquiring of other skills, such as the abilities of excellent communication, business management, and successful problem solving. These programs should focus on developing the capability of businesses to grow, become resilient, and achieve the growth target.

Every country defines the significance and role of the "Green Economy" according to their own vision, and every country defines it with the aim of realizing its vision.

The Serbian strategy for a green growth economy is based on:

(1) Knowledge-based sustainability;
(2) Socioeconomic conditions and perspectives;
(3) Environment and natural resources.

The Republic of Serbia has 57 laws and 32 strategies that are directly or indirectly connected to sustainable development, environment, green economy, and green growth (Mineral Resources Management Strategy of the Republic of Serbia until 2030; The National Strategy for the Inclusion of the Republic of Serbia in the Clean Development Mechanism of the Kyoto Protocol for the sectors of waste management, agriculture, and forestry; The Strategy of Cleaner Production in the Republic of Serbia; The National Strategy for Approximation in the Environmental Field for the Republic of Serbia; The Law on Environmental Protection, and so forth) [36].

The Ministry of Agriculture of Serbia has published the Rulebook, which sets out the conditions for obtaining incentives worth up to three million dinars for the development of rural tourism and old artistic crafts. In addition, Program for allocation of funds from the EU Instrument for Pre-accession Assistance for Rural Development (IPARD) programs can be effective tools in encouraging business in rural tourism and agriculture growth.

The tourism development strategy in Serbia for the period 2016–2025 emphasizes the importance of fostering tourism, not only through new value creation and through employment, but also through

the multiplier effects that tourism has on local and regional development, development of culture and education, and improvement of women's lives. The objectives of this strategy are [24]:

- Sustainable tourism development,
- Strengthening the competitiveness of the tourism industry,
- Increasing tourism's share of GDP and employment,
- Improving the image of the Republic of Serbia in the global tourism market.

Women's entrepreneurship development is possible within the activities involved in different types of tourism (hospitality, accommodation, gastronomy) as well as within activities relating to it. These primarily include old crafts, homecraft (making handicrafts), trade, processing of products, etc. Women's entrepreneurship also involves the creation of new tourism services that are imperative for the development of the offerings of modern tourism [29].

Kolmogorov test variables were studied according to the *t*-Test and Pearson Chi-Square correlation analysis after determining null and first hypotheses of normality. Our research has indicated that women mostly prefer specialized programs of education that are adjusted to the job requirements. The research has also indicated that working skills themselves do not exclusively empower women. So, it showed that hypothesis H0 is not confirmed. The hypothesis H1 is confirmed because our study showed that education is not critical for success in business. This was even the opinion of those among the respondents with the highest education (56%). This means that higher education institutions do not offer enough practical knowledge.

The research stressed that women lacked the assistance of the state to start up their firms, and very often, they do not have financial support from the state or their families. Namely, most of the problems concerning the implementation of green economy in SMEs is a lack of financial support, directly or indirectly. Even with the existence of different national and international programs, financing of investments in green entrepreneurship in tourism still represents a great problem, which occurs not only in SMEs owned by women, but also in other types of companies.

Lastly, how education will help in empowering women depends on close relationships between government, educational institutions, and women's organizations.

This research paper contributes to a theoretical approach by reviewing the literature on sustainable tourism development and green entrepreneurship, as well as by reflecting on the women's empowerment. Further research is required to develop appropriate financial instruments that fit the needs of these small tourism enterprises. These should include investment subsidies, tax measures, and better offers of lending products.

## 9. Recommendation

Ensuring that SMEs fully participate in the efforts towards green growth in Serbia is a key challenge for the transformation ahead. In line with the concept, the Green Economy policies should be designed to reflect long-term social, economic, and environmental public interests, with an emphasis on the SME development, reduction of poverty, and employment growth [6,47]. In other words, the Green Economy policy can be supported by clear processes for integrating environmental, social, and economic goals, along with national strategies for implementing goals across responsibility areas.

In this context, we can recommend an appropriate policy that can be most effective in empowering women in tourism industry. The policy should include the following measures:

- Constantly training and educating employees about the importance of sustainability and "green thinking ";
- Promoting women's education and training according to job needs in tourism;
- Achieving full coverage of the territory of the country with support services in training and education for entrepreneurship in tourism;
- Providing lower taxes and special credit lines for women who want to start up their own businesses;

- Increase awareness of the potential entrepreneurs and those who want to set up their own companies in the tourism industry;
- Promoting women's entrepreneurship and facilitating equal access to start-up grants;
- Faster application of the Law on Environmental Protection and changing Environmental Legislative according to SMEs needs, etc.

**Author Contributions:** M.R.-M. made the conceptual framework for the study, did all statistical analyses, provided data, wrote the paper, and revised it several times. B.Ž. conducted the interviews in banks and supervised the research. All authors reviewed the manuscript several times.

**Funding:** This research received no external funding

**Acknowledgments:** The research was supported by the Ministry of Education, Science and Technological Development, the Republic of Serbia (Grant III 47009 and 179015).

**Conflicts of Interest:** The authors declare no conflict of interest.

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
