# Peer review of "Fostering Green Entrepreneurship and Women’s Empowerment through Education and Banks’ Investments in Tourism: Evidence from Serbia"

_sustainability, doi:10.3390/su11236826_

Round 1

Reviewer 1 Report

Dear Author(s)

Please find here with my comments, belief will be helpful to revise your manuscript...

In abstract author(s) mentioned gender...it is suggested to use women instead of gender... The title of study is "Fostering Green Entrepreneurship and Women  Empowerment through Banks' Investments in Tourism: an Evidence of Serbia"....No where in the abstract I find how Banks' Investments in Tourism foster green entrepreneurship and women empowerment.... Introduction: Introduction section need to be more rigorous...it fails to motivate what is need of present study... Introduction author quoted "However, the growth of tourists reflects on the environment.".....please mention few references.....this is the heart of the study... In introduction section authors quoted UNWTO studies conclusion in 4 points...but what is missing is how these points are related to their study.....may these serves as the motivation for the present study... Introduction author(s) quoted two objectives but there is no objective on how Banks' Investments in Tourism foster green entrepreneurship and women empowerment. So study title need to be changed...or re-framed on the line of present research... Hypotheses framed on just imagination...I strongly suggest the author(s) to discuss the concerned literature before framing such hypothesis....  Theoretical overview: Author(s) quoted "Numerous schools"....please specify them... Theoretical overview section is very immaturely written need more serious and comprehensive discussions.  Method of the study: Please specify the type of sampling used...How 120 respondent selected ???  Please Provide the descriptive statistics of the data that justifies the quality of the data... Key findings and discussion: Author(s) quoted “Do you have formal 119 education in tourism? “ what did they mean by formal education is it university degree or what basis not clear??? Overall Key findings and discussion section is not well organised and need serious discussions...which is missing.....Author(s) suggested to compare their findings with previous studies results...... Opportunities for strengthening the position of women in tourism section is not back by any established theories.... Conclusion and recommendation section: Author(s) quoted "Our research indicated that the largest number of female respondents in all age groups.....".....how ??? provide evidences.... Author(s) are suggested to avoid this kind of statements..."hypothesis H1 is partly confirmed"..... Overall conclusions are too generic readers know it like "the biggest problem they face when starting a business in tourism is a lack of financial resources and then lack of appropriate skills..." English need to be improved References need to be rechecked as many cases page, volume and issue numbers are missing. Final suggestion to the author(s) to make the study empirically strong they may consider SEM to derive study conclusions...

Good Luck !!!

Author Response

Dear Reviewer,

Thanks a lot for useful suggestions.They are accepted and according them revision is made.

Please,see the attachment.

Thanks again!

Best,

Authors

Reviewer 2 Report

Dear authors,
First of all, I would like to say that your article is very interesting, but it is necessary that you make a very important effort so that it can be published.
Here are some general comments about it and some bibliographical references that can help you improve it.

1-Introduction
The introduction should specify that we know about Green entrepreneurship and women empowerment to date, because it is interesting to study this topic, how it contributes to the development of research, what is the gap that is intended to be covered and what are the objectives of the work.
2. Hypotheses
The hypotheses must arise from a previous theoretical approach. You should review articles on the subject to rework it. The hypothesis is the end of a deduction process, not the beginning.
3. Theory and methods
It must be connected with the hypothesis and the work.
The methods are not described correctly. It only appears that a questionnaire was made, but how was the information processed, how was the results reached? Was the questionnaire validated?

Results
A series of descriptive graphs does not seem the appropriate methodology for a magazine such as Sustainability. Regression analysis, correlation, structural equation models, in short, something more solid from a methodological point of view.

Suggestion

Keywords: ODS, Tourism inclusive, sustainable developmet goals

Some articles to reference

Ferreira Gregorio, V.; Pié, L.; Terceño, A. A Systematic Literature Review of Bio, Green and Circular Economy Trends in Publications in the Field of Economics and Business Management. Sustainability 2018, 10, 4232.

Martín, J. M. M., Aguilera, J. D. D. J., & Moreno, V. M. (2014). Impacts of seasonality on environmental sustainability in the tourism sector based on destination type: An application to Spain's Andalusia region. Tourism Economics, 20(1), 123-142.

Khoshnava, S.M.; Rostami, R.; Zin, R.M.; Štreimikienė, D.; Yousefpour, A.; Strielkowski, W.; Mardani, A. Aligning the Criteria of Green Economy (GE) and Sustainable Development Goals (SDGs) to Implement Sustainable Development. Sustainability 2019, 11, 4615.

Núñez-Cacho, P.; Molina-Moreno, V.; Corpas-Iglesias, F.A.; Cortés-García, F.J. Family Businesses Transitioning to a Circular Economy Model: The Case of “Mercadona”. Sustainability 2018, 10, 538

Author Response

Dear Reviewer,

Thanks for excellent suggestions.They are accepted and revision is done according them.I hope that my hard work on article improvement is visible.

All the best,

Mirjana

Round 2

Reviewer 1 Report

Dear Author(s),

I have gone through your revised work...I am not totally satisfied with the improvements. I can see only few of my suggestions are incorporated...So I hope this time you will take my previous comments seriously and act upon it. Please make a table of two columns put heading reviewer comments and authors comments.....and address all the comments made one by one next time. 

Good Luck !!!

Author Response

Dear Reviewer,

I am sending in red marked what is changed or added new in the text.

Mirjana

Reviewer 2 Report

Dear Authors,   Firstly, I have noticed that you have made some improvements to your work.  I ask you to make an extra effort so that this article can be published.  I believe that you must  incorporate all the bibliographical references that I have provided you in the first revision so that they reinforce this article.  
Thank you very much for your effort

Author Response

Dear Reviewer,

I marked in red what is changed or added in the text. I made a major revision according to your suggestions.

Thanks a lot for useful comments.

Mirjana

//Editor's comments: the authors later provided a file listing the comments and the responses, so we removed the former attachment (which was the manuscript file).

Round 3

Reviewer 1 Report

Dear Author(s),

I have gone through your modified manuscript...Point number 3 to 11 are not addressed at the satisfactory level. It need serious revision:
For example:
Point number 3: You mentioned that "Studies almost have shown the role of women in their workplace, but fewer their position in the tourism sector"...Which is not true, there are many studies done with role of women in tourism sector...Just type it in google scholar you can find many...but it is strongly suggested to search Scopus or EBESCO or similar databases.    Then check for point no. 4: I ask for few reference for this quoted line "However, the growth of tourists reflects on the environment." References that author(s) cited is not enough.....Using these references you have to discuss "how growth of tourists reflects on the environment" from the quoted studies...it is missing     Point no. 7: Author(s) quoted "Numerous Schools"...I ask to specific them...Author(s) have to justify which schools: classical, behavioral...... with appropriate justifications...   Similarly other points are not addressed in satisfactory manner as expected... I belief now author(s) will take these opportunity as final and serious to address all the concerned specially as mentioned in points no. 3 to 11.   Good Luck !!!

Author Response

Dear Reviewer,

I did the third revision of my paper with the aim to improve it according to  all your proposals. So, I made corrections and entered 8 new references in the text which supported theoretical part. Also, I added 7 pages more in my article. Namely ,while the first version had 11 pages , the latest one after the third revision has 18 pages.

I would like to point out that all parts of the text are improved with new explanations ,comments ,tables and etc. Also ,conclusion is revised and written again. All changes are marked this time in blue ( first version was in black ,second in red but third in blue).

Best,

Mirjana

Reviewer 2 Report

Dear authors,   I have verified that they are incorporating improvements in their article, but I believe that there is still room for improvement so that it can be published and in the future it can be a reference article in this field.  In this sense, I would like to point out again some articles that contain information and ideas to strengthen the introductory part, methodology and conclusions.   I would like to point out some of these articles for your consideration.   I want to thank you for your effort in improving your research work.   Martín, J. M. M., Aguilera, J. D. D. J., & Moreno, V. M. (2014). Impacts of seasonality on environmental sustainability in the tourism sector based on destination type: An application to Spain's Andalusia region. Tourism Economics, 20(1), 123-142   Khoshnava, S.M.; Rostami, R.; Zin, R.M.; Štreimikienė, D.; Yousefpour, A.; Strielkowski, W.; Mardani, A. Aligning the Criteria of Green Economy (GE) and Sustainable Development Goals (SDGs) to Implement Sustainable Development. Sustainability 2019, 11, 4615.

Author Response

Dear reviewer,

Thanks again for your suggestions. I did my best to improve my article in all its parts. After third revision my article increased the number of pages ,i.e.  first version before revision had 11 pages ,but the third version has 18 pages. In other words, 7 pages are added. All new entered in the text is marked this time in blue.

I hope that you will be at last satisfy.

Kind regards,

Mirjana

Round 4

Reviewer 1 Report

Dear Author(s),
Thank you for the revised manuscripts...This time significant improvement has been incorporated from your side. Still few things are left out and I belief you will incorporate this:

1. Point no. 6 need more detailed discussions : Discuss studies in details and based on it formulate your hypothesis. 

"Hypotheses framed on just imagination...I
strongly suggest the author(s) to discuss the
concerned literature before framing such
hypothesis"

2. Point no. 9 is not addressed: Provide Mean, standard deviation, skewness, kurtosis, corelations among the variables.

"Please Provide the descriptive statistics of the
data that justifies the quality of the data.."

3. Paper needs a professional english proof reading.

Good Luck !!! 

Author Response

Dear reviewer ,

I did my best for the fourth time to make some improvements according to your suggestions. In green is marked what is new added in this version. I hope the final one  because other reviewer has accepted the paper for publication yet. Also, the manuscript was sending for English editing. I sent proof of professional English editing to the Journal editor in chief to show you.

Thanks for your time and useful proposals.

Please,see the attachment.

Kind regards,

Mirjana

Reviewer 2 Report

Dear Authors,   After having reviewed the article, I consider that the changes that have been introduced have made it possible for this article to be published.     I encourage you to continue working in this field of knowledge.

Author Response

Dear Reviewer,

Thank you very much for nice words and useful suggestions.

All the best!

Mirjana